



**Spatiotemporal Evaluation of Vertical Dynamics Propagation of Flash**
**Drought and Driving Mechanisms in the Indus Basin in South Asia (1970-**
**2023)**
**Tahira Khurshid[1], Qiongfang Li[1,2], Chuanhao Wu[2], Akif Rahim[3], Muhammad Shafeeque[4,5,6], Shanshui**
**Yuan[2], Zia Ul Hassan[7], Junliang Jin[2,8,9]**
1College of Hydrology and Water Resources, Hohai University, 210098 Nanjing, China
2The National Key Laboratory of Water Disaster Prevention, Hohai University, 210098 Nanjing, China
3International Water Management Institute, 53700 Lahore, Pakistan
4Climate Lab, Institute of Geography, University of Bremen, 28359 Bremen Germany
5Alfred Wegener Institute (AWI), Helmholtz Centre for Polar and Marine Research, 27570 Bremerhaven, Germany
6MARUM - Center for Marine Environmental Sciences, University of Bremen, 28359 Bremen Germany
7Department of Hydraulic Engineering, Tsinghua University, 100190 Beijing, China
8Yangtze Institute for Conservation and Development, Nanjing 210098, China
9Research Centre for Climate Change of Ministry of Water Resources, Nanjing 210029, China
Corresponding authors: Qiongfang Li (qfli@hhu.edu.cn) & Chuanhao Wu (wuch0907@hotmail.com)

## Abstract

Flash drought (FD) leads to relatively short periods of anomalously low and rapid
decreasing soil moisture (*SM*), which can significantly affect vegetation growth and ecosystem.
However, the vertical propagation characteristics and driving mechanisms of FD through soil
columns remain largely unknown, which is crucial for guiding agricultural and ecological disaster
prevention and reduction. Here, we present a multi-layer FD evaluation method to explore the
vertical propagation of FD at different soil depths (0-10 cm, 10-40 cm, and 100 cm), and based on
the GLDAS data we comprehensively evaluate the spatiotemporal dynamics and driving
mechanisms of FD during 1970-2023 in the Indus Basin, a highly climate-sensitive region. We
find that the frequency of FD decreases with increasing soil depth, while the relationship between
the rate of intensification (*RI*) and drought severity varies with soil depth, with stronger correlation
($r^2>0.9$) in the middle and root zone soil layers than in the upper layer. We further identify 2148
simultaneous events ('*t*') and 1154 subsequent events ('*t*+1') between the upper and middle soil
layers, which mostly occur during spring and early summer. The temporal differences in the '*t*+1'
FD events are closely related to the persistence of meteorological conditions. In contrast, the '*t*'
events are caused by the simultaneous depletion of *SM* from the upper layer to the deeper layer,
indicating the rapid development of FD conditions due to deeper moisture loss. The analysis also
highlights the significant spatial heterogeneity of FD characteristics, with the humid and sub-
humid regions in the middle Indus basin being the most sensitive to FD, and precipitation deficit
and high temperature are the dominant driving forces for FD occurrence.





**Key words:** Flash drought, rate of intensification, soil moisture anomalies, vertical propagation,
GLDAS, Indus basin.

## 1 Introduction

Drought is a complex hydrometeorological phenomenon with far-reaching impacts on
agriculture, water resources, and ecosystems (Mishra et al., 2010). Among various drought types,
flash drought (FD) has emerged as a critical area of concern due to its rapid onset and potentially
severe consequences (JaSOn et al., 2018; Osman et al., 2020; Yuan et al., 2023). Unlike
conventional droughts that develop over months or years, FD can manifest within weeks, leaving
little time for preparation and mitigation (Pendergrass et al., 2020; Ford and Labosier, 2017). For
example, a severe FD affected the central Great Plains and Midwest regions of the United States
(US) in 2012, causing agricultural losses exceeding \$30 billion (Otkin et al., 2019). The FD that
occurred in 2017 resulted in agricultural losses of \$2.6 billion in the Dakotas and Montana, U.S.
(Dilling et al., 2019). FD occurs when precipitation ($P$) deficit is accompanied by above-average
evaporative demand due to high temperatures, increased vapor pressure deficit, low humidity, and
strong winds (Otkin et al., 2019, 2016). If such a condition persists for days to weeks, it will force
the system to transition from an energy-limited state to a water-limited state, resulting in a rapid
increase in soil moisture (SM) deficit and the development of FD in a region (Ford and Labosier,
2017; Hunt et al., 2014, 2009; Možný et al., 2012).
*SM* anomaly is a useful indicator for characterizing the onset of FD (Hunt et al., 2009;
Možný et al., 2012). The rapid depletion of *SM* is one of the first signs of FD since evaporative
demand directly contributes to *SM* depletion (Otkin et al., 2016; Ford and Labosier, 2017; Osman
et al., 2020; Yuan et al., 2019). *SM* anomalies appear first in topsoil before moving deeper into the
soil (Hunt et al., 2014; Ford et al., 2015). Decreased *SM* in the upper layer stimulates the vegetation
roots to take up water from deeper soil depths (Baudena et al., 2013). Across the globe, significant
advances have been made in exploring the changes of FD events at different *SM* layers (Shah et
al., 2022; Zhang et al., 2022; Sungmin and Park, 2023) and in root zone soil moisture (RZSM) as
a particular focus (Lesinger et al., 2022; Mahto and Mishra, 2023; Zheng et al., 2022). These
valuable studies address multiple aspects of FD, including meteorological drivers, rate of
intensification (*RI*), ecological impacts, and future assessment. However, the interaction between
*SM* layers regarding FD vertical propagation has not been assessed before. It is unclear at what
time lag FD events of the upper layer tend to propagate deeper soil layers in response to the
meteorological anomalies. Evaluating the temporal dynamics of FD transmission across soil layers
and related meteorological conditions enhances our understanding of drought mechanics, aiding
local preparedness and response plans. In addition, there is no research focusing on the influence
of *RI* on FD severity at various soil depths. It remains unknown whether the impacts of *RI* on
drought severity are dependent on soil depth.
FD poses unique challenges for water resource management and agricultural planning,
particularly in regions with high climate variability and strong land-atmosphere coupling (Basara
et al., 2019a; Mahto and Mishra, 2020). The Indus Basin, spanning parts of Pakistan, India, China,



and Afghanistan, is a transboundary basin with diverse climatic zones, ranging from arid lowlands
to snow-capped mountains, supporting the livelihoods of over 268 million people (Laghari et al.,
2012; Shafeeque et al., 2022). The water resources in this basin are already under stress due to
population growth, agricultural intensification, and climate change (Immerzeel et al., 2016;
Shafeeque et al., 2023; Shafeeque and Bibi, 2023), making it particularly vulnerable to rapid onset
drought events. Despite the significant impacts of FD events, there is a notable gap in
understanding the spatiotemporal dynamics and driving mechanisms of FD in the Indus Basin.
Although several studies have investigated long-term drought patterns in the Indus Basin (Adnan
et al., 2018; Ashraf et al., 2023; Ashraf et al., 2021; Abbas et al., 2021), research specifically
focusing on FD remains limited. Aside from being studied in some valuable global-scale research
(Neelam and Hain, 2024; Mahto and Mishra, 2023; Mukherjee et al., 2022; Sreeparvathy et al.,
2022; Qing et al., 2022; Yuan et al., 2019; 2023), the in-depth investigation of FD characteristics
in the Indus Basin is one of the crucial gaps in the literature. This gap is particularly concerning
given the projected increases in climate variability and extreme events in this region (Lutz et al.,
2016; Shafeeque and Bibi, 2023; Krishnan et al., 2019).

Here, based on the *SM* and meteorological data from the Global Land Data Assimilation
System (GLDAS) over the period 1970-2023, this study presents a multi-layer FD evaluation
method to explore the temporal evolution of FD from upper (0-10 cm) to middle (10-40 cm) and
RZSM layer (100 cm) in the Indus Basin, a typical water-limited region, to improve our
understanding of FD vertical propagation. Based on the timings of occurrences of FD events, we
explore the simultaneous and subsequent FD events between different *SM* layers and compare the
associated meteorological conditions to determine potential differences. Specifically, our study
aims to address the following scientific questions: 1) how do the dynamics of FD vary across
different soil depths in the Indus Basin, particularly regarding the spatio-temporal distribution of
FD and the impact of *RI* on FD severity, 2) what is the tendency of the FD events to vertically
propagate between different soil layers and the role of meteorological anomalies in the FD
development? 3) what is the evolutionary behavior of meteorological variables before and after
FD? and how do these variables influence *RI*?
This study contributes to the growing body of literature on FD by examining the vertical
propagation of FD through the soil column, offering insights into their three-dimensional nature.
Additionally, it provides valuable insights for improving drought monitoring, early warning
systems, and water management strategies in the Indus Basin and similar complex river basins
worldwide.

## 110   2 Methodology

### 111   2.1 Study area

The study domain covers the Indus Basin of South Asia, which is located at 22°–38°N and
65°–87°E (Fig. 1). The basin encompasses a total land area of 1.12 million km$^2$, shared by four
adjacent countries namely Pakistan (47%), India (39%), China (8%), and Afghanistan (6%). It



stretches from the western Himalayan-Karakoram-Hindu Kush Mountains in the north to the dry
alluvial plains of Pakistan in the south. The Indus Basin has a unique climate, with a humid- semi
humid climate in the north and a semi-arid to arid climate in the south (Hasson et al., 2019;
Shamsudduha et al., 2019). The annual average precipitation in the basin is about 230 mm (Janjua
et al., 2021), but the precipitation pattern is irregular, with pronounced variability in magnitude,
time of occurrence, and aerial distribution (Ahmad et al., 2014). Temperature varies from below
freezing (<0°C) at higher elevations during winter to above 40°C at lower elevations during
summer (Krakuer et al., 2019). The mean annual potential evaporation ranges from 1650 to 2040
mm (Ali, 2013; FAO, 2011), and is expected to increase with global warming (Janjua et al., 2021).
In the past decades, this region has witnessed numerous severe and extreme drought events
(1984–86, 1992, 1998–2002, 2007/08, 2012/13, 2017/18, and 2022) (Rahman et al., 2023), making
it a critical area for study due to the significant drought impacts. For example, an extreme FD event
occurred unexpectedly in late spring in 2022 and continued throughout the summer across
Pakistan, with the temperature exceeding 51°C in some parts of the country during May (Nanditha
et al., 2023), leading to reduced wheat yield, livestock losses and health issues (Otto et al., 2022).
This region is vulnerable to more frequent and intense drought events due to climate warming,
which has significantly influenced food and water security (Ali et al., 2022).

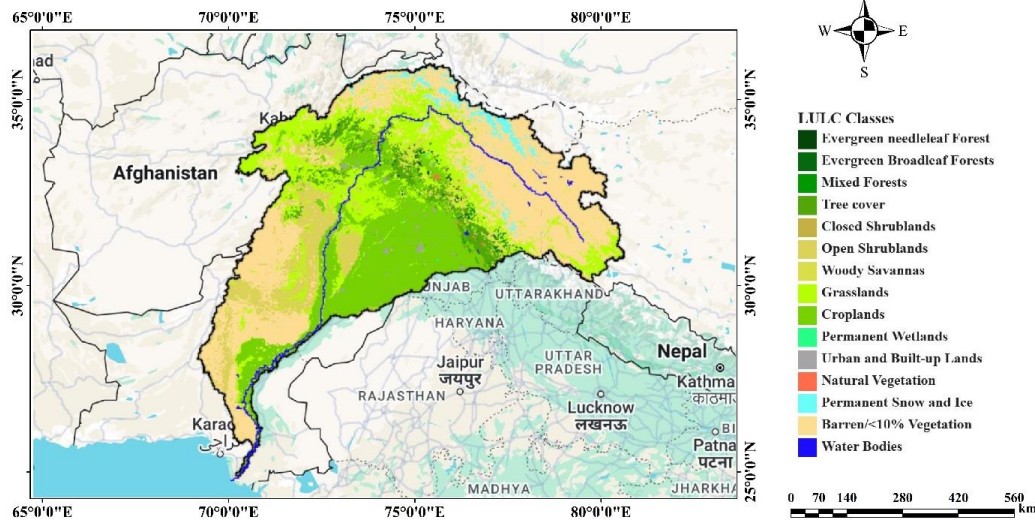


**Figure 1.** Study area map (Indus Basin) showing vegetation classes, derived from the MODIS dataset
(https://code.earthengine.google.com/scriptPath/MODIS/061/MCD12Q1).
**2.2 Data collection and processing**
To quantify FD events and explore the associated meteorological driving mechanisms over the
Indus Basin, the 8-daily data of *SM* at different depths (0-10 cm, 10-40 cm, 100 cm) and
meteorological variables including precipitation (*P*), temperature (*T*), evapotranspiration (*ET*), and



wind speed (*WS*) during 1970-2023 were collected from the GLDAS Version 2 (GLDAS-2) dataset
with a horizontal resolution of $0.25° \times 0.25°$. GLDAS-2 is reprocessed with the updated Princeton
Global Meteorological Forcing Dataset (Sheffield et al., 2006) and upgraded Land Information
System Version 7 (LIS-7). This dataset has been proven to effectively capture *SM* trends and
patterns at both regional and global scales (Jia et al., 2018; Dorigo et al., 2012; Cheng et al., 2015).
In this study, we construct an 8-daily average time series of all input variables to avoid high-
frequency variability in hourly data.

### 2.3 Identification of FD at various soil depths

In this study, FD is defined based on *SM* at the 8-daily scale. We employ an 8-daily time
scale, as it effectively captures temporal variations in hydrometeorological variables, and well
reflects the rapid severity of FD as well as their impact on agroecosystem (Hu et al.,2025; Yang et
al., 2023). We identify FD events at three different *SM* depths, i.e., 0-10 cm, 10-40 cm, and 100
cm, which directly affect the growth status of vegetation.
As *ET* is limited in winter, which prevents the rapid decline of *SM*, this study focuses on
the growing season (April to October) and also includes the FD events that begin in March and
end in April (Christian et al., 2020). An FD event occurs when the following conditions are met:
1.    When the 8-daily mean *SM* (%) decreases from above the $40^{th}$ percentile to the $20^{th}$
percentile within 3-time steps (i.e., 24 days), it indicates the "onset" stage of FD. The
$20^{th}$ percentile of *SM* is selected as the drought threshold according to Yuan et al. (2019).

2.    The average *RI* of *SM* is not less than 5% in the percentile during the onset of FD.
$$R\,\mathrm{Im}\,ean = \frac{1}{n}\sum_{1=0}^{i}[\frac{SM(t_{i+1})-SM(t_i)}{t_{i+1}-t_i}] \geq 5^{th}percentile \tag{1}$$

3.    After the onset phase, the average *SM* of the next three octads remains below 20%. The
recovery stage of FD starts when the *SM* percentile begins to increase. Once the *SM*
recovers to above the $20^{th}$ percentile, the drought ends.



**Figure 2.** Illustration of the identification of FD.

Spatiotemporal characteristics of FD used in this study include the number of events, intensity, and severity. The severity refers to the accumulated *SM* percentile deficits from the 40% threshold (shaded area in Fig. 2) (Yuan et al., 2019). Intensity denotes the average *RI* during the drought onset period. The study period (1970-2023) is divided into two halves (1970-1996 and 1997-2023) to examine the evolutionary variations in FD characteristics over time.

We conduct the correlation analysis between *RI* and drought severity to explore the impacts of *SM* depth on FD characteristics at various soil depths.

In addition, we apply the one-way analysis of variance (ANOVA) to test the sensitivity of the impact of *RI* on drought severity to different *SM* layers. The ANOVA is a commonly used statistical hypothesis tool to evaluate the variations between two or more groups of observations (Hattermann et al., 2018). We constructed the null and alternate hypotheses as:

$H_O$ = the impact of *RI* on drought severity is the same in all *SM* layers.

$H_1$ = the impact of *RI* on drought severity varies with respect to the depth.

The significance of any variations is determined by the *F-test* at the significance of 0.05. The *F-test* is recommended as a practical test, because of its robustness against many alternative distributions (Hattermann et al., 2018). The *F*-statistics explains the degree to which the variability in *RI* that can be attributed to the difference between soil layers is proportionately larger than the variability within each soil layer.





## 2.4 Evaluation of Vertical propagation of FD events

The evolution of key meteorological factors (*P, T, ET, and WS*) directly affects the development of FD events. These driving factors can either confine the FD events to the upper layer or facilitate their penetration into deeper layers simultaneously or subsequently. In this study, we evaluate the vertical propagation of FD events between different layers (i.e., upper and middle) by comparing the two types of FD events: i) that occur in both layers at the same time ('*t*-events', i.e., simultaneous events), and ii) that occur in the middle layer at 1-lead time '*t*+1' time to the upper layer ('*t*+1 events', i.e., subsequent events). Further, meteorological anomalies are used to assess the difference in meteorological variations responsible for these two types of FD events. The monthly distributions of these FD events from March to October are also compared and evaluated to determine their probable timing of occurrence.

## 2.5 Analysis of driving forces of FD

The meteorological anomalies play an essential role in driving the development and recovery of FD events (Otkin et al., 2014). To understand the meteorological conditions associated with the FD occurrence, we evaluate the variability of *T*, *P*, *ET*, and *WS* anomalies before and during FD events at the 8-daily scale during the study period (1970-2023), which are calculated as follows:

$$Anomaly_t = \frac{x_y - \bar{x}}{\sigma}$$

(2

where '$x_y$' is the actual 8-day average meteorological factor (*T*, *P*, *ET*, and *WS*) in '*t*' time steps for '*y*' year, '$\bar{x}$' is the mean 8-day average meteorological factor in '*t*' time steps for the study period, '*Anomaly$_t$*' is the anomaly in '*t*' time steps for '*y*' year.

The anomalies of *T*, *P*, *ET*, and *WS* are calculated for each grid cell across the study area. We evaluate their evolutionary behavior with two-time steps back (lag-2, lag-1) and forth (Lag+1, lag+2) and during the onset of FD events (lag0) using the boxplot distribution.

The multiple linear regression (MLR) model is further used to quantify the relative contribution of meteorological variables (*T, P, ET,* and *WS* ) to *RI* (Ting et al., 2009; Cheng et al., 2015). The equation of MLR is expressed as follows:

$$RI_i = \alpha_0 + \alpha_1 X_{1i} + \dots + \alpha_n X_{ni} \left( i = 1, 2, \dots, m \right)$$

$$RI = \begin{bmatrix} RI_1 \\ RI_2 \\ . \\ . \\ RI_m \end{bmatrix}, \quad X = \begin{bmatrix} X_{11} & \dots & X_{n1} \\ X_{12} & \dots & X_{n2} \\ . & . & . \\ . & . & . \\ X_{1m} & \dots & X_{nm} \end{bmatrix}, \quad \alpha = \begin{bmatrix} \alpha_0 \\ \alpha_1 \\ . \\ . \\ \alpha_n \end{bmatrix}$$

(3



where $X_i = P_i$, $T_i$, $ET_i$, and $WS_i$ are meteorological anomalies during the $i$th drought event; $\alpha_0$ and
$\alpha_{1-n}$ are intercept and regression coefficients, respectively; $m$ indicates the number of events at a
given grid cell; and $RI_i$ represents the regressed rate of intensification at a given grid cell estimated
by the MLR model. A regression coefficient reflects the importance of an independent variable to
a dependent variable (Zhang et al., 2022). Using regression coefficients, we can determine the
relative importance of meteorological variables on $RI$. The performance of the MLR model is
evaluated based on the coefficient of determination ($R^2$):

(4

$$R^2 = \left( 1 - \frac{\sum_{i=1}^{n} \left( RI_{obs}(i) - RI_{smi} \right)}{\sum_{i=1}^{n} \left( RI_{obs}(i) - \overline{RI_{obs}} \right)^2} \right)$$

where $RI_{obs}(i)$ and $RI_{smi}(i)$ are observed and simulated $RI$ at the grid '$i$', respectively; $\overline{RIobs}$ mean
observed $RI$, $n$ indicates the number of observations.
**3. Results**
**3.1 FD characteristics at different SM layers**

The maximum number of FD events during 1970-2023 is recorded in the upper layer (i.e.,
44) demonstrated by the widespread spatial pattern of events across the basin (Fig. 3a), because it
tends to respond quickly to atmospheric variability and interacts closely with the evaporative
demand (Potkay et al., 2021; Jafari et al., 2021). However, we observe an obvious shift in the
pattern of occurrence from the lower to middle basin during the first and second half of the study
period (Figs. 3b & c). In the case of the middle layer, the number of FD events varies between 1-
33 throughout the basin. The middle Indus basin experiences a maximum number of events (~ 21)
between 1997 - 2023, which is revealed by the spatial extent from the east to the west and the
southwestern region of the study area (Fig. 3f). Further, the analysis of RZSM shows that the
increased spatial extent of events occurrences in the second half (1997-2023) is consistent with
that of the other two layers, with the number of events ranging from 1 to 39. Generally, the middle
basin located in the transition zone from humid to semi-humid and semi-arid is more prone to FD
events.



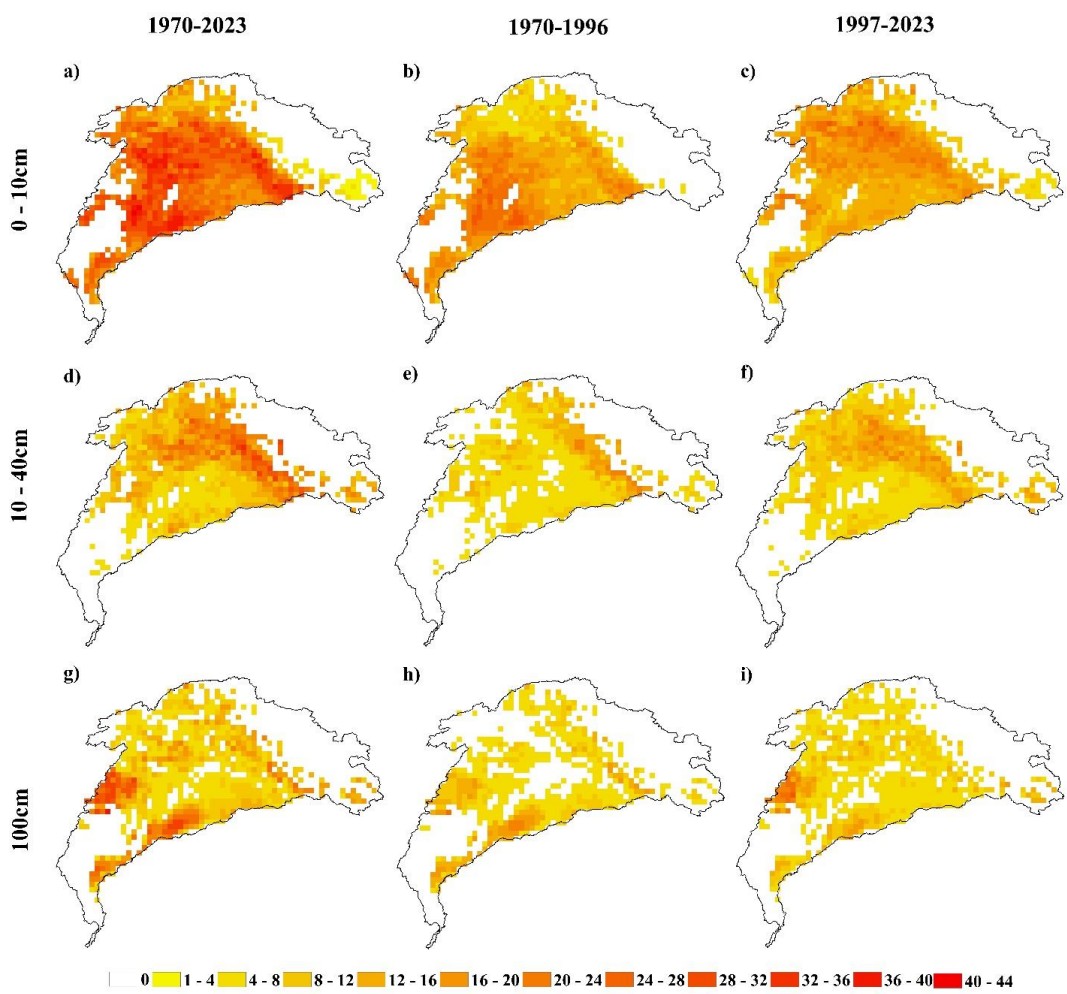

**Figure 3.** Spatial distributions of the number of FD events at three *SM* layers (0-10 cm, 10-40 cm, 100 cm) during the periods 1970-2023, 1970-1996, and 1997-2023.

Similarly, Figure 4 demonstrates the spatiotemporal pattern of the *RI* of the total FD events over the study period (1970-2023). The *RI* describes the average depletion rate of *SM* during the onset-development of FD (Xiang et al., 2020). According to Fig. 4, the spatial distribution of *RI* is consistent with the number of events at each layer. In general, a higher *RI* (i.e., 55 percentile) is observed in the middle basin, while a lower *RI* (i.e., < 5 percentile) is found in the southern part of the basin.

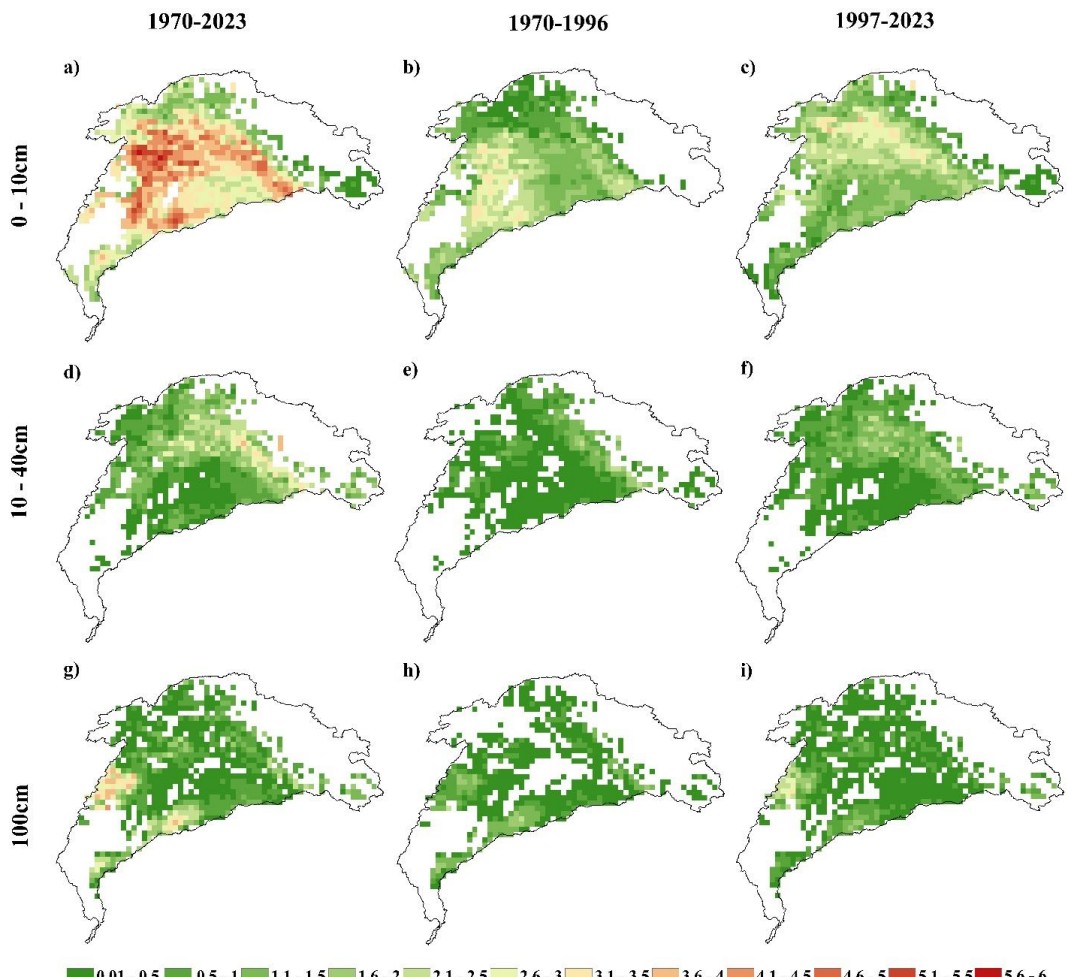

**Figure 4.** Spatial distributions of the average *RI* of the total FD events at three *SM* layers (0-10 cm, 10-40 cm, 100 cm) during the periods 1970-2023, 1970-1996 and 1997-2023

Fig. 5 shows the total accumulated severity of FD events over the Indus Basin during 1970-2023. In all three layers, the spatial pattern of severity is similar to the number of FD events at the grid scale. The middle Indus Basin experiences a higher severity than other regions. During 1997-23, we observe a noticeable variation in spatial extent. A significant rise in the number of events with higher severity is observed in the upper and the middle layers from the east to the west of the basin during 1997-2023 (see Fig. 5c & f). Similarly, the drought severity in the RZSM layer during 1997-2023 (Fig. 5i) is higher in the western region of the basin compared to 1970-1996 (Fig. 5h).

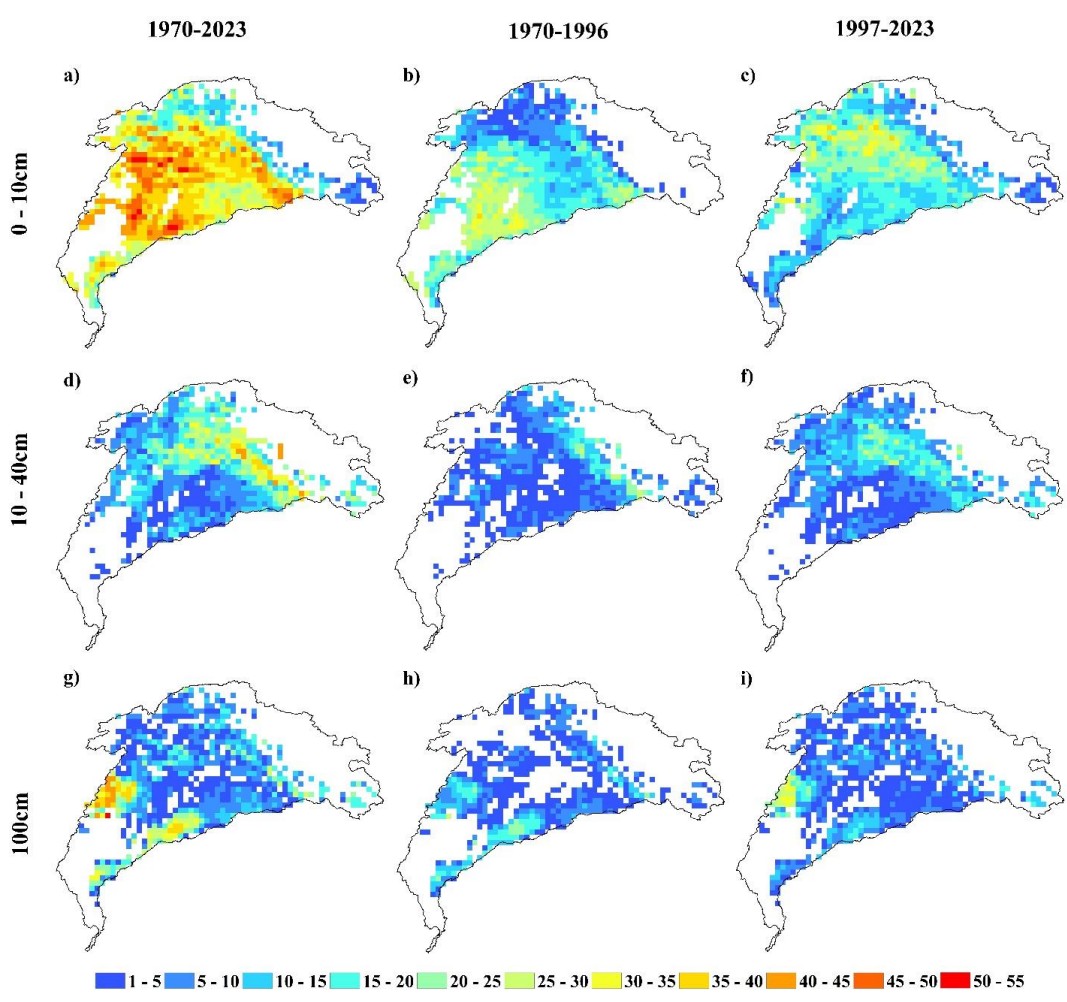

**Figure 5.** Spatial distributions of the total accumulated severity of FD events at three *SM* layers (0-10 cm, 10-40 cm, 100 cm) during the periods 1970-2023, 1970-1996 and 1997-2023

Fig. 6 demonstrates the spatial distributions of the correlation between *RI* and drought severity. In the upper layer, there is a weak correlation ($r^2$<0.35) between *RI* and severity over 94% of grids (Fig. 6a), suggesting that the *RI* shows little impact on drought severity in the upper layer. Comparatively, the middle layer exhibits a moderate to strong correlation ($0.40 < r^2 < 1.0$) between *RI* and severity in some parts of the basin (Fig. 6b), especially in Punjab province (highlighted with a red circle, $r^2$>0.9), which is covered by flat alluvial land and has extensive agricultural activities (Abbas et al., 2013; Kumar et al., 2013). Similarly, a strong correlation is found in RZSM (Fig. 6c), with the $r^2$ ranging from 0.4 to 1 across the middle Indus Basin, especially in Punjab province (encircled with red color). This may be due to the fact that the vegetation facilitates



drawing more water from deep soil during the drought condition to fulfill the evaporative demand
(Wang et al., 2016). Moreover, the strength of the correlation can be attributed to the persistency
of *SM* at different soil depths, which is supported by the analysis of ANOVA, showing that the
difference between the layers is statistically significant ($F = 18.0193$, $p < 0.05$). It is noteworthy that
there is a weak correlation ($r^2 < 0.2$) in the south-western part of the basin (blue dotted area) (Fig.
6c), where more frequent FD events are detected (see Fig. 3g). It can be attributed to the low
moisture content in this part of the basin, which leads to a weak response to high evaporative
demand, resulting in a poor correlation between *RI* and FD severity. Hence, the above analyses
suggest that the middle Indus Basin exhibits a significant influence of *RI* on severity. Moreover,
the spatial heterogeneity of *SM* influences drought severity.

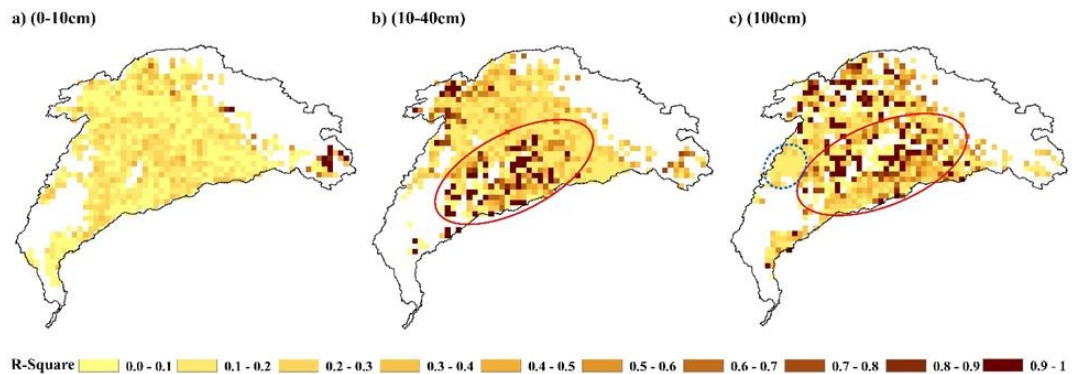


**Figure 6.** Spatial distributions of the correlation coefficients ($r^2$) between *RI* and drought severity for each *SM* layer.
**3.2 Vertical propagation of FD events across the SM layers**
We identify 2148 '*t*-events' and 1154 '*t*+1 events' during 1970-2023 (see Fig. 7e). Both
types of events are spatially distributed across the middle Indus Basin and some parts of the upper
Indus Basin (Figs. 7a & c), where overall climate varies from humid to semi-humid, supporting
vegetation growth and agricultural activities. However, the spatial extent of the '*t*-events' is greater
than the '*t*+1 event'. As '*t*-events' are characterized by the immediate and severe *SM* deficits from
the upper to the middle *SM* layer that can cause severe risk to the ecosystem. While, in the case of
the '*t*+1 event', when the decline in *SM* in the middle layer occurs at the one-time step after the
decrease in the upper layer, it suggests a slightly slower progression of dryness from the upper to
deeper layer. These events can exacerbate the drought impacts initiated by the upper layer, as the
prolonged dry conditions in the middle layer can lead to extended periods of drought stress for
vegetation. This transferring impression of FD from the upper to the deeper layer is in line with
the existing literature, which indicated that with the development of FD, the *SM* deficit first appears
in the upper moisture layer, and then leads to further downward movement of the deficit to the
deeper layers (Otkin et al., 2016; Anderson et al., 2013). The finding of the vertical FD propagation
highlights the importance of *SM* monitoring at various depths in drought monitoring systems.




Figs 7b & d explain the monthly distribution of FD events between March to October. September witnessed the maximum number of both types of events throughout the study period (1970-2023) followed by April for 't-events', and May for 't+1 events'. Moreover, for the 't+1 events', the number of events gradually increases from March to May (Fig. 7d), indicating an increased likelihood of the FD occurrence from spring to early summer. The emergence of 't-events' and 't+1 events' during these months are in line with the FD event that occurred in the US Northern High Plains in 2017, where warm-dry weather during spring and early summer caused a severe drought over two months (Otkin et al., 2019). Similarly, a large number of FD events in Spain were recorded during spring and summer (Noguera et al., 2020).

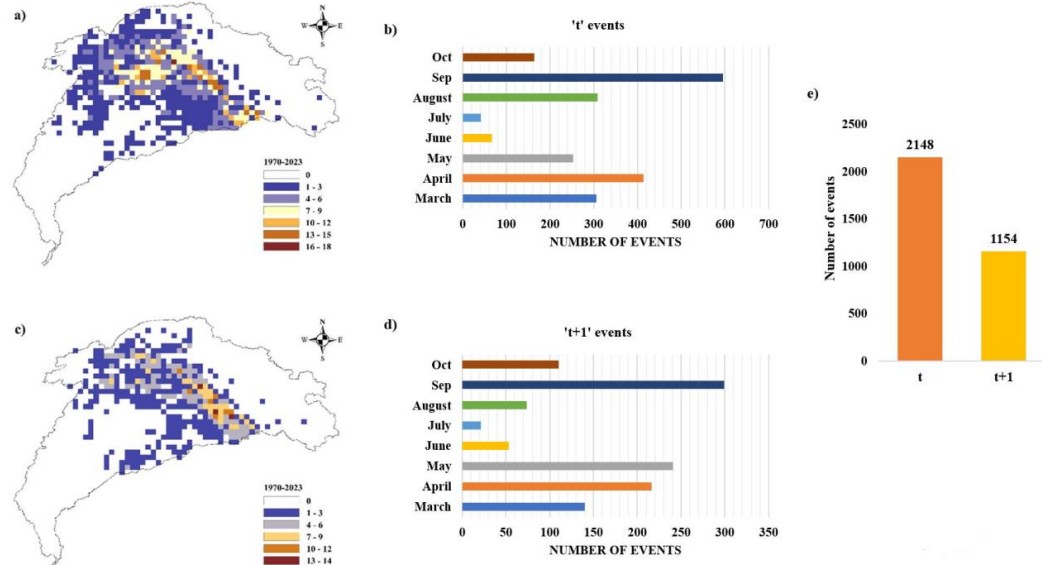

**Figure 7.** Vertical propagation of FD events between different *SM* layers. (a) shows the events in both layers at the same time '*t*', and (c) events in the deep layer at 1 lead time '*t+1*' time with respect to the upper layer. b) and d) explain the monthly distribution of '*t*' and '*t+1*' events, respectively. e) shows the total number of '*t*' and '*t+1*' events.

### 3.3 Meteorological driving mechanism of FD

Based on the correlation results in 4.1 section, we use the RZSM to assess the meteorological evolutionary behavior associated with FD events. Fig. 8 illustrates the boxplot distribution of the anomalies of four meteorological variables before and after the FD events at each time step (lag-2 to lag+2). The higher maximum $T$ anomalies are > 2 standard deviations (STDs). The distribution pattern for each time step (lag-2 to lag+2) shows that the median of $T$ anomalies is lower at lag-1, but becomes stronger at the onset (lag0) and after the onset of FD (Fig. 8a).



Unlike *T* anomalies, the median of *P* anomalies shows a decreasing trend among all
temporal lags (i.e., lag-2 to lag+2, Fig. 8b), with the maximum negative anomaly of -1.5 STDs.
There is an obvious decrease in maximum positive *P* anomalies from lag-2 to lag-1, indicating the
development of conditions leading to drought occurrence. From lag-1 to lag+2, the maximum *P*
anomalies tend to show a negative STD (i.e., > -1 STD), with the median around -0.5 STD. Overall,
the *P* anomalies remain negative before and during the onset of FD events, and this behavior
persists as drought progresses. It is important to note that significant negative *P* anomalies may
result in an increase in *T* that rapidly dries out *SM*. The depletion of *SM* due to the negative *P*
anomalies could lead to a decrease in *ET*, leading to higher *T* and vapor pressure deficits (VPDs).
A higher VPD further accelerates the *SM* depletion (Zhou et al., 2019). Moreover, the negative *P*
anomalies accompanied by the positive *T* anomalies can intensify the atmospheric water demand,
thus further speeding up *SM* decline and triggering FD in a region (Mahto and Mishra, 2020).
The distribution pattern of *ET* anomalies shows a shift from negative (lag-2) to positive
anomalies (lag-1 and lag0) (Fig. 8c), which coincides with positive *T* and negative *P* anomalies
during lag-1 and lag0 (see Figs. 8a & b). The median of *ET* anomalies becomes more negative (>
-2 STD) as the drought proceeds (from lag+1 to lag+2), which indicates the worsening of the
drought condition. In this situation, the high *T* increases evaporative demand, leading to a system
from energy-limited to water-limited states. The distribution pattern of *WS* does not reveal the
distinguished variations before and after FD (Fig. 8d). The higher maximum positive *WS* is
observed at lag0 (> 3 STDs), while the median remains positive and slightly tends to increase from
lag-1 to lag+2. All in all, the variations in the behavior of *T*, *P*, *ET*, and *WS* during different stages
of FD indicate a close association between FD development and meteorological variabilities in
this region, which is consistent with the previous findings (Mo et al., 2016; Yuan et al., 2019).

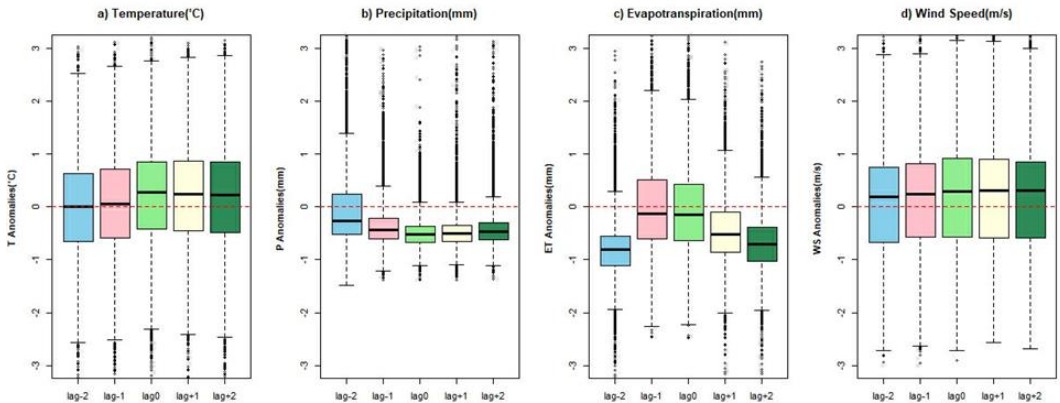

**Figure 8.** Boxplots of the standardized anomaly values of meteorological variables in adjacent time steps of FD events at the
RZSM layer (100cm) during 1970-2023 extracted from all grids over the Indus Basin. Lag-2, and lag-1 show time steps prior to
the FD onset, lag0 represents FD onset time, and lag+1 and lag+2 show the time steps after the onset of FD.



346    Further, to examine meteorological variabilities associated with the '*t*' and '*t*+1' FD events,
347  we used a common factor anomalies approach mentioned above. Regarding the meteorological
348  variabilities associated with '*t*-events', we observe an increase in the median of *T* from lag-1 to
349  lag+2 (Fig. 9i(a)), with the highest positive *T* deviation at lag+1. In the case of *P*, the median
350  decreases from lag-1 to lag+1, and the maximum *P* deficit continues until lag+2 (Fig. 9i(b)). This
351  coevolution ultimately reduces the humidity level of the system, which appears from the *ET*
352  variability from lag0 to lag+2 (Fig. 9i(c)). However, for *WS*, there is a slight increase in median
353  from lag0 to lag+1, suggesting an increase in positive anomalies after the onset of FD.

354    Compared with '*t*-events', we observe drastic *P* deficits and high *T* at lag-1 for '*t*+1 events'
355  (Fig. 9ii (a & b)). These FD events evolve from the events that appeared at the upper layer, hence
356  inheriting the particular impression of meteorological variabilities that can sustain drought
357  characteristics. The maximum *T* anomalies are > 2.5 STDs from lag-1 to lag+2, with the positive
358  median during this time frame (Fig. 9ii(a)). We notice persistent negative *P* anomalies from lag-1
359  to lag+1 (Fig. 9ii(b)), with the values slightly increasing from lag+1 to lag+2 but still below 0. On
360  the other hand, the median of *ET* anomalies gradually decreases from lag-2 to lag+2 (Fig. 9ii(c)),
361  which coincides with the variability of *T* and *P* (Fig. 9ii (a & b)). The increase in *T* and decrease
362  in *P* from lag-2 to lag+2 substantially decrease *SM*, thereby reducing *ET*.



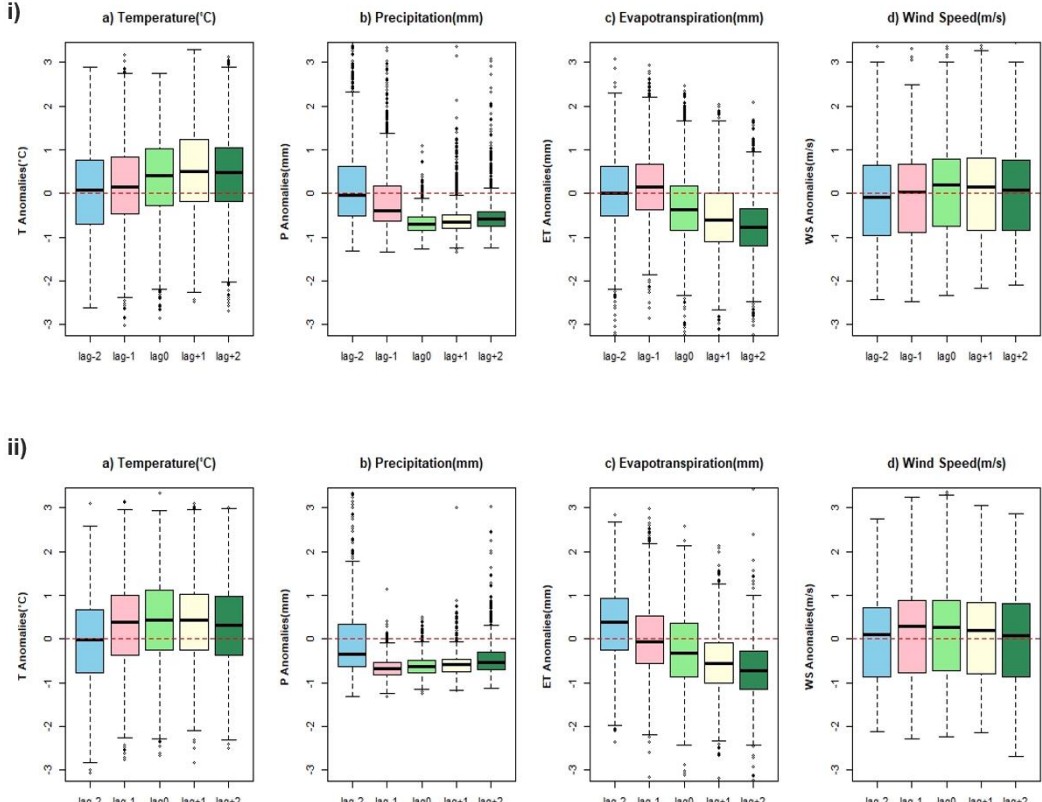

**Figure 9.** Same as in Figure 8, but for (i) '*t*-events' and (ii) '*t*+1 events'.

Figure 10a demonstrates the weights of meteorological anomalies contributing to *RI* during the onset development phase of FD events. On average, we observe the largest negative weight of *P* anomalies, suggesting the dominant role of *P* deficit in the FD onset in this domain. *T* and *ET* present nearly equal contributions to *RI*, which is in line with Koster et al. (2019), stated that the *P* deficit is the main driver of the FD occurrence. The model demonstrates a strong coefficient of determination ($R^2 > 0.9$) in the middle Indus Basin (see Fig. 10b), suggesting a significant contribution of meteorological variables to *RI*. In contrast, a weak $R^2$ (~ 0.4) is found in the southern region of the basin. Overall, the spatial pattern of $R^2$ is in harmony with the findings in Fig. 6, suggesting that the middle Indus Basin is more sensitive to FD occurrence compared to other regions.



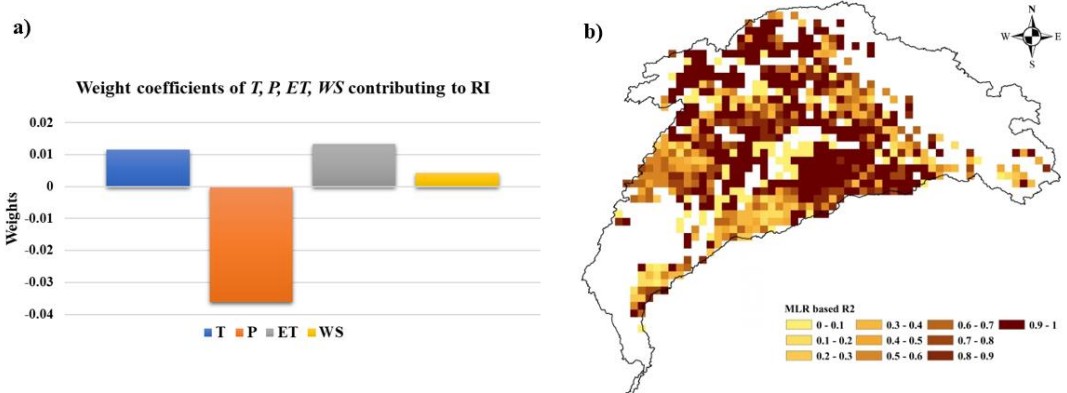

**Figure 10.** a) Weight coefficients of meteorological variables (*T, P, ET, WS*) contributing to *RI* during the development phase of
FD. b) Spatial distribution of the coefficient of determination ($R^2$) of the model fitted by the multiple linear regression (MLR).

## 4. Discussions

### 4.1 Impact of SM dynamics on FD characteristics

This study finds the occurrence of FD events at each *SM* layer. However, the FD frequency
is much larger in the upper layer compared to the deeper layer, mainly because the *SM* within the
upper layer is most sensitive to meteorological variability, allowing more events to be detected.
The results are consistent with a previous study on California (Cheng et al., 2016), which reported
a higher drought frequency within the upper *SM* layer. We observe a distinctive increase in the
spatial extent of FD events in the second half (1997-2023) of the study period (Figs. 3c, f, and i).
It is important to discuss here that anthropogenic-driven climate warming has changed
considerably around 2000 (Zeng et al., 2023). Recent studies have shown a dramatic increase in
atmospheric water demand since 2000, leading to more frequent occurrence of heatwaves and
other related disturbances including *P* and *SM* deficits (Williams et al., 2020; Perkins-Kirkpatrick
et al., 2020). Our findings are in good agreement with Ullah et al. (2024), reporting the
intensification in FD frequency and severity in Sri Lanka, Afghanistan, Pakistan, and India during
2010-2020. Moreover, the results highlight the tendency of the humid and semi-humid regions to
induce FD, which has also been appreciated in previous research (Basara et al., 2019b). During
1997-2023, the number of FD events is noticeably amplified in humid and semi-humid regions of
the middle Indus Basin, which leads to a higher variability in *SM* dynamics and results in more
frequent drier and wetter anomalous conditions in such regions. Shah et al. (2022) reported a high
occurrence of FD in humid regions. Chen et al. (2021) also reported a higher possibility of the FD
occurrence in regions that have higher *SM* variability.

There are various vegetation types (crops, grassland, and forests) in the middle Indus basin
due to the humid to semi-humid climate (Fig. 1). Such regions are likely to sustain *ET* in response
to higher *T* as long as vegetation roots make water available from the system to fulfill the
evaporative demand (Christian et al., 2019b; Zhu et al., 2021), hence developing the favorable



conditions for the FD occurrence. This is in accordance with Guo et al. (2023), which demonstrated
the role of vegetation water consumption in the onset of FD in humid regions. Zeng et al. (2023)
also reported an increase in FD frequency in humid regions of the Amazon basin and Congo basin.
Similar findings were noted in Russia where FD occurred in agricultural areas including Central,
Volga, and southern federal districts of southwestern Russia in 2010 due to $P$ deficit (Christian et
al., 2020). In contrast in arid regions, higher $T$ inhibits $ET$ due to the limited $SM$ availability (Mo
et al., 2016; Yuan et al., 2019), resulting in a lower likelihood of FD occurrence than in humid
regions (Wang et al., 2016).
The relationship between $RI$ and drought severity is very sensitive to the spatial and vertical
variability in $SM$, and a stronger correlation ($r^2 > 0.7$) is found in the middle and RZSM layers
compared to the upper layer. This indicates that the impact of $RI$ on drought severity varies with
soil depth. This variability across the vertical $SM$ column apprehends a more holistic dynamics of
$SM$ variation (Berg et al., 2017). Also, regarding the spatial extent, the correlation between $RI$ and
drought severity is stronger in the regions with higher $SM$ level. Consequently, moisture from the
system is depleted considerably in response to the higher $RI$.
As significant agriculture activities are carried out in the middle Indus Basin, the
occurrence of FD events can pose substantial impacts on vegetation productivity. Hence, it is
evident that the frequency of FD events alone does not dictate the severity of drought impacts on
the system. The other factors, such as $SM$ deficit at different depths, $RI$, and land characteristics
also play critical roles in exacerbating drought severity. In conclusion, we can say the FD events
that occurred in the middle and RZSM layers can produce significant impacts on agroecosystem
and water resources, and the varying effects of $RI$ at various soil moisture layers can pose a
potential concern for different types of vegetation.

## 4.2 Influence of land-atmosphere interaction on the FD vertical propagation

The soil water content in the vertical soil profile plays an important role in the soil-plant-
atmosphere continuum system (Western et al., 2004). The assessment of the vertical propagation
of FD events between different $SM$ layers shows that the middle Indus Basin and some lower parts
of the upper Indus Basin are more susceptible to FD. Given the combined effect of $P$ deficit and
high $T$ (Figs. 9i & ii), the amount of $ET$ could increase in a short period ultimately leading to the
development of simultaneous '$t$' and subsequent '$t$+1' FD events. The increase in transpiration
during the growing season exacerbates the rapid decline in $SM$ (Koirala et al., 2017). According
to Qing et al. (2023), the increasing drying trend of soil water in humid regions corresponds to
positive $T$ and negative $P$ anomalies. This deficit of moisture within the surface, sub-surface soil,
and atmosphere ultimately results in a drier vertical atmosphere over a region (Dimri et al., 2019).
However, this pattern cannot be acquired in arid regions, because sparse or no vegetation reduces
$ET$ due to limited $SM$ supply ( Qing et al., 2022; Gupta et al., 2020).
This is the first study investigating the temporal relationship in FD occurrence across
different $SM$ layers. The analysis of the '$t$+1' events reveals that the relationship between different



*SM* layers is influenced by variations in the moisture content of the upper *SM* layer, which
facilitates the vertical propagation of FD. The temporal differences in the occurrence of '*t*+1' FD
events can be associated with the persistence of certain meteorological conditions in a region that
drive the vertical depletion of *SM* over time (Fig. 9ii). Moreover, the loss of moisture from different
soil depths is affected by different factors. For example, soil evaporation and transpiration mainly
consume the water in the upper soil layer, whereas water loss in the deeper soil layers depends on
transpiration, which leads to different responses to meteorological anomalies at various time lags.

On the other hand, the '*t*' events result from the simultaneous depletion of *SM* from the
upper layer to the deeper layer, indicating the rapid development of FD conditions due to deeper
moisture loss, and reflecting the dynamic behavior of the FD phenomenon over the same region.
Overall, the study provides new insights into how the varying moisture conditions at different soil
depths affect the occurrence and dynamics of FD events that can be important for FD monitoring
and prediction systems for a region.

The results also indicated that the '*t*' and '*t+1*' events occur most frequently in spring and
early summer (Figs. 7b & d). During this season, a dipolar system develops over southern Asia
expanding from east to west, which actively propagates the pressure gradient and adiabatic wind,
greater condensation, and *P* deficit, thereby amplifying the FD occurrence in this region (Ullah et
al., 2024). Moreover, vegetation growth can also affect the development of FD. As warming *T*
advances the spring leaves' emergence (Fu et al., 2015), the plant growth in the early stages may
reduce *SM* in late spring due to enhanced *ET* (Angert et al., 2005; Peñuelas et al., 2001). The *SM*
deficits may continue till summer due to further increased *T* and lead to more severe FD. This may
explain the occurrence of FD in vegetative-dominant areas, particularly during spring and summer
(Figs. 7a & c). The spatial extent of these extreme FD events is relatively less compared to the
total extent of all FD events recorded in each *SM* layer (Fig. 3). Xu et al. (2021) also found that
the areas affected by extreme drought decrease with the increase of soil depth during the most
severe stage of drought. This may be related to the differences in the persistency property of *SM*
at different depths.

### 468   4.3 Contribution of meteorological anomalies to the FD occurrence

The rapid onset of FD is attributed to local moisture imbalance caused by anomalies in *P*
and *T*. The simultaneous occurrence of meteorological anomalies may cause rapid development of
FD by decreasing *SM* (Basara et al., 2019b; Burrows et al., 2019; Fischer et al., 2007; Soares et
al., 2019; Qing et al., 2022). We analyzed the variations in meteorological variables (*T*, *P*, *ET*, and
*WS*) at different lags (lag-2 to lag+2) before and after FD to explore the underlying mechanism of
FD development. We observe negative *P* and *ET* anomalies and positive *T* anomalies during the
FD onset phase (Fig. 8). The negative *ET* anomalies correspond to the low *SM* during FD (Osman
et al., 2020). The consistent *P* deficits can cause a decrease in *ET* leading to a higher *T* and vapor
pressure deficit, which contributes to the onsets of heatwave and *SM* deficiency. This type of FD
is considered a precipitation deficit FD (Zhou et al., 2019). The decrease in *ET* is also the result of
rising *T* in a region (Zhang et al., 2017). The FD in Russia developed in July and mid-August in




2010 due to *P* deficit and previously desiccated land surface, and led to a rapid rise in surface *T*
and vapor pressure deficit. Christian et al. (2020) suggested the impact of FD as a potential
precursor to heatwaves.

Fig. 8 shows that the meteorological anomalies persist as the drought proceeds, which
suggests that the Indus Basin is more prone to precipitation deficit FD. Similarly, the occurrence
of FD in the neighboring country of India during the monsoon season may be associated with the
delay or reduction in monsoon rainfall (Christian et al., 2020; Mahto and Mishra, 2020). However,
*WS* anomalies do not show obvious variations before and after the FD occurrence (Fig.8 & 9),
which is consistent with the existing studies (Liu et al., 2020; Gou et al., 2021; Li et al., 2022)
reporting no significant positive or negative *WS* anomalies during FD.

We also investigate the impacts of meteorological variables on *RI*, which is an important
characteristic that distinguishes FD from conventional drought (Otkin et al., 2019). The
meteorological forcing of FD is stronger than that of conventional drought (Ford and Labosier,
2017), which explains the close interaction between *RI* and meteorological conditions. The MLR
analysis highlights the dominant role of *P* in influencing *RI*, followed closely by *ET* and *T*. The
model demonstrates a strong coefficient of determination ($R^2 > 0.9$) in the middle Indus Basin and
semiarid region of Punjab province, which is sensitive to the interaction of land and atmosphere.
For example, a decrease in *SM* can lead to a decrease in *ET* and limit the local sources of boundary
layer moisture, ultimately reducing atmospheric moisture advection. Subsequently, the atmosphere
remains dry, increasing evaporation demand, and the dry soil is not conducive to convective
rainfall, thereby exacerbating FD (Basara et al., 2019a, 2019b; Christian et al., 2019).

Overall, our study highlights the importance of the early warning systems being
reconstructed to incorporate the FD phenomenon based on the understanding of the mechanism
involved in FD development. FD is more likely to develop in humid and sub-humid regions, with
more severe impacts on ecosystems and agriculture. Therefore, developing a framework involving
improvement in the early warning system, preparedness, and modification in existing strategies
can help prepare a proactive system and minimize the expected losses.

## 5. Conclusion
This study provides the first comprehensive analysis of FD dynamics in the Indus Basin,
revealing complex spatiotemporal patterns and vertical propagation characteristics. The results
indicate a higher occurrence of FD events in the middle Indus Basin in 1997-2023 compared to
1970-1996, suggesting an increased risk of FD over time in this region. The FD events are more
prone to occur in the humid and sub-humid regions of the middle Indus Basin. Their frequency
tends to decrease from the upper to the RZSM layer, with the strongest correlation between *RI* and
severity ($r^2 > 0.9$) in the middle and RZSM layers, which is linked to the persistency of *SM*.

The vertical propagation analysis suggests that the susceptibility of the region to FD is
driven by rapid loss of deeper *SM* ('*t*') and slower depletion of *SM* from upper to deep soil ('*t*+1').
Moreover, the number of '*t*' events and their spatial extent are greater than that of '*t*+1' events.



There is a close relationship between meteorological conditions (rapid $T$ rise and $P$ decline) and FD development, especially in the '$t$+1 events' at lag-1. The temporal differences in the '$t$+1' FD events are closely related to the persistence of meteorological conditions. In contrast, the '$t$' events are caused by the simultaneous depletion of $SM$ from the upper layer to the deeper layer. The MLR analysis further suggests that $P$ is the most contributing factor affecting $RI$, followed closely by $T$.

These findings have significant implications for drought management and early warning systems in the Indus Basin, informing more effective agricultural and water management practices. Future research will focus on the transition from FD to conventional drought, investigating the impacts of climate and land cover changes, and developing coupled land-atmosphere models to improve FD prediction.

## Appendix A: List of abbreviations used in this study

| FD | Flash Drought |
|---|---|
| SM | Soil Moisture |
| RZSM | Root zone soil moisture |
| RI | Rate of Intensification |
| T | Temperature |
| P | Precipitation |
| ET | Evapotranspiration |
| WS | Wind speed |
| MLR | Multiple linear regression |

## Data Availability

The dataset used in this research is publicly available at Google Earth Engine and can be accessed from the following link: http://earthengine.google.com/.

## Author Contribution

Tahira Khurshid developed the methodology, conceptualization, data collection, performed the analysis, writing and original draft preparation, Qiongfang Li supervision, funding acquisition, review and editing, Chuanhao Wu funding acquisition, review, editing and refinement of draft, Akif Rahim supported for data acquisition and FD identification, Muhammad Shafeeque Writing, review and editing, Shanshui Yuan, Zia Ul Hassan, Junliang Jin review and editing.






## Competing interests

The authors declare that they have no conflict of interest.

## Financial Support

This work is supported by the National Natural Science Foundation Commission of China (Grant
number 524034911).

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
