# Peer review of "Spatiotemporal Evaluation of Vertical Dynamics Propagation of Flash Drought and Driving Mechanisms in the Indus Basin in South Asia (1970-2023)"

_EGUsphere, 2025_

## Author Comment (AC1)

**Spatiotemporal Evaluation of Vertical Dynamics Propagation of Flash Drought and Driving Mechanisms in the Indus Basin in South Asia (1970-2023)**

Tahira Khurshid[1], Qiongfang Li[1,2], Chuanhao Wu[2], Akif Rahim[3], Muhammad Shafeeque[4,5,6], Shanshui Yuan[2], Zia Ul Hassan[7], Junliang Jin[2,8,9]

**RC1: General comments:**

This manuscript investigates the vertical propagation of flash drought in the Indus Basin, South Asia, over the period 1970-2023. The topic is interesting, and the manuscript is generally well-written. However, the overall quality needs improvement. The current results do not sufficiently illustrate the driving mechanisms behind vertical flash drought propagation. Several critical issues related to datasets, methodology, and results require further clarification. Therefore, I recommend major revisions for this manuscript.

**Specific comments:**

1) The GLDAS dataset includes multiple versions with different spatial and temporal resolutions, such as GLDAS-NOAH, GLDAS-CLSM, and GLDAS-VIC. The authors should explicitly specify which dataset was used.
   Response
   Thank you for the comments. In this study, GLDAS-NOAH 3hrly dataset has been employed to investigate the flash drought. 3hrly dataset was converted into 8-daily data to prevent the high-frequency variability in flash drought assessment. Following your comment, we will incorporate this information in revised MS.

2) The authors employ 0-10 cm, 10-40 cm and 100 cm soil depths for flash drought identification. However, the meaning of 100 cm layer is unclear. Does it refer to 40-100 cm and 0-100 cm? Additionally, the authors state their intention to explore vertical flash drought propagation from upper layer to the root-zone soil moisture (RZSM) layer. But RZSM is commonly defined as the 0-100 cm soil layer, which appears inconsistent with previous definition of middle layer (10-40 cm).
   Response
   Thank you for the comments. We are sorry for unclear information.
   a) The 100cm soil moisture layer refers to '0-100cm'. The relevant text in the document will be revised to better clarify about the depth of the RZSM layer considered in this study.
   b) The vertical flash drought propagation has been investigated on two soil moisture layers (0-10cm and 10-40cm), as mentioned in the methodology section 2.4 (see L187-190). 0-100cm layer is not included in this part. Because it describes the average soil moisture conditions in soil column from 0-100cm, and therefore couldn't explain the occurrence of flash drought in relation to different depths. According to your comment, L24 in the

Abstract section will be improved to remove the confusion and enhance the clarity about this point.

3) The rationale behind selecting the four variables for flash drought analysis needs to be elaborated. I am confused why wind speed is included, given that it does not directly induce a rapid increase in soil moisture. On the other hand, some critical hydrometeorological variables such as potential evapotranspiration, vapor pressure deficit, and solar radiation should be considered, as they are commonly associated with flash drought occurrence and development.

Response

Thank you for the comments.

   a) **Rationale behind selecting meteorological parameters:** The meteorological variables examined in this study were selected based on established findings from prior research. For instance, Deng et al. (2022) demonstrated that flash droughts (FD) are usually accompanied by insufficient precipitation (P) and evapotranspiration (ET). Similarly, Qing et al. (2022) highlighted that high temperature (T) and precipitation deficits develop flash drought occurrences globally. In addition, vapor pressure deficit (VPD) incorporates relative humidity and temperature (Mahto and Mishra, 2023), and can show a high correlation with temperature (Feng et al., 2024). Given that temperature is already a focal variable in our investigation of FD mechanisms, VPD was excluded from the original analysis.

   b) Regarding wind speed, several studies (e.g., Li et al., 2022; Otkin et al.,2019, 2016) stated the contribution of strong winds along with other hydrometeorological variables in FD occurrence. Consequently, the present study evaluated whether wind speed significantly contributes to FD occurrence in the study region, and we found its weak role in the FD occurrence in the Indus Basin.

   c) Following your suggestions, the other three hydrometeorological variables (VPD, SR, PET) will be considered in the revised MS to assess their role in FD occurrence in this region.

4) Figure 3 presents inconsistencies in flash drought occurrences across different soil layers. For example, subplot (h) suggests that the number of flash drought events at 100 cm is highest in the southern areas, exceeding 32 events. However, in panel (e), the number of events in the 10-40 cm is below 20 for the same areas. This implies cases where deeper soil moisture depletion occurs while the middle soil layer remains relatively unaffected, which seems counterintuitive. Similar inconsistencies are also observed in panels (f) and (i).

Response

Thank you for the comments.

The southern area of the basin is characterized by arid and semi-arid climate (Waseem et al., 2020; Adams, 2019) with sparse and erratic rainfall (Waseem et al., 2020) and limited and seasonally dependent surface water resources (Ashraf and Ul hasaan, 2019). The previous study (Dimri et al., 2019) indicates that high temperature and evapotranspiration lead to the drying of available moisture within the surface and subsurface soil in this part of the region. The detection of flash drought in deeper soil layer can be attributed to the availability of soil moisture at this depth, while soil moisture in the upper and middle layers is rather limited due to stronger evapotranspiration. Therefore, the upper and middle layers are more difficult to meet the criteria

for flash drought caused by rapid decline in soil moisture compared to deeper soil layer, leading to more frequent flash drought in deeper soil layer. Besides that, the soil texture and structure also have an evident effect on the variation of moisture at different soil depths (Ajmal et al., 2016), though this factor is not explicitly examined in the present study but remains a focus of ongoing research. Moreover, agriculture in the arid Indus Basin is sustained through water extraction from the groundwater and Karez system (Ashraf and Ul hasaan, 2019), which can also affect the vertical distribution of soil moisture across different soil layers.

According to your comment, we will make a detailed discussion in the revised MS to further support our findings.

5) The study primarily focuses on the severity of flash droughts. However, it is equally important to examine their duration, as severity is closely linked to drought durations. The authors should provide an analysis of flash drought duration across different soil layers. Does the upper soil layer experience shorter flash drought duration than deeper layers? Is this result relevant to the weak correlations observed in Figure 6 between RI and severity in the upper soil layer?

Response

Thank you for the comments.

Analysis in Figure 6 considers the severity over the next three consecutive time-steps (i.e., 24 days) following the onset phase of flash drought. The total duration of flash drought is not accounted for, as differences in drought duration may lead to higher soil moisture deficits in flash drought with prolonged duration. To assess the influence of soil depth variability on flash drought characteristics, a fixed threshold was applied across all three soil moisture layers. This criterion ensures consistency in evaluating the impact of RI on the severity of flash drought. This detail will be incorporated into the revised MS to clarify the temporal framework used in this analysis.

Following your suggestions, we will analyze and compare the flash drought duration across different soil layers.

6) The definition of vertical flash drought propagation process remains unclear. The classification of "t" events appears to indicate the simultaneous occurrences of flash droughts in different soil layers, but this does not necessarily imply propagation. Additional evidence is needed to demonstrate that upper-layer flash droughts propagate into deeper layers in the so-called "t" and "t+1" events.

Response

Thank you for the comments.

In this study, the vertical flash drought propagation process indicates the downward movement of drying conditions through soil layers. Specifically, moisture deficits in the upper layer during a FD event induce deficit in middle layer, leading to FD occurrence at that depth. We identified two types of the events i.e., simultaneous ('*t*') and subsequent ('*t*+1'), based on the criteria mentioned in section 2.4 (see L187-190). '*t*' events signify the quicker response of middle layer to upper layer, while in case of '*t*+1' events, middle layer response was observed at the difference of 1-timestep (i.e., 8-days). The drought propagation is characterized by the difference in the FD onset time between upper and middle soil moisture layers.

Following your suggestion, additional evidences explaining the relationship of soil drying in the upper and middle layer will be incorporated into the revised MS.

7) The current results (Figure 9) do not convincingly demonstrate the influence of hydrometeorological variables on the occurrence of "t" and "t+1" events. More results, similar to Figure 10, should be included to identify the critical variables driving these events and to illustrate the underlying mechanisms of vertical flash drought propagation.

Response

Thank you for the comments. Following your suggestion, we will compute weight coefficients of meteorological variables contributing to RI during "t" and "t+1" events, respectively, and spatial distribution of the coefficient of determination following Figure 10 to better identify the critical meteorological variables associated with the occurrence of "t" and "t+1" events.

8) In Figure 8, the variations in precipitation from lag-2 to lag0 appear smaller than those of evapotranspiration and temperature, suggesting that ET and T may exhibit stronger correlations with flash drought occurrence. However, this contrasts with the findings in Figure 10, where precipitation is identified as the key driver of flash droughts. The authors should clarify this inconsistency.

Response

Thank you for the comments.

Figure 8 is somewhat different from Figure 10, which displays the contributions of meteorological variables to RI during the onset-**development phase (from lag0 to lag+2)** of flash drought.

In addition, although the range of precipitation variability from lag-2 to lag0 is smaller than those of evapotranspiration and temperature variability, the average abnormal deviation (~ median value) of precipitation is larger than that of evapotranspiration and temperature, especially from lag-1 to lag0. This suggests that precipitation shows an overall larger contribution to the occurrence of flash drought than evapotranspiration and temperature, which is generally consistent with Figure 10.

According to your comments, we will provide relevant information in the revised MS.

9) I suggest using (a)-(h) for subplot labels instead of "i)"and "ii)"in Figure 9.

Response

Thank you for the comments. Following your suggestion, we will use (a)-(h) for subplot labels instead of "i)"and "ii)"in Figure 9.

10) It is unclear which soil layer is represented in Figure 10. 0~10 cm, 10~40 cm, or 100 cm? Additionally, why does the study not analyze the driving factors for flash droughts across all three soil layers? It would be informative to determine whether the dominant factors remain consistent across different soil depths.

Response

Thank you for the comments. We are sorry for unclear information about Figure 10. In this figure. soil layer represented is 0-100cm. We will provide relevant explanation in Figure 10 in the revised MS. In addition, following your suggestion, we will analyze the driving factors for flash droughts across all three soil layers to explore the discrepancy in the dominant factors of flash drought across different soil depths.

**Refrences:**

Adams III, T. E.: Water resources forecasting within the Indus River Basin: A call for comprehensive modeling, In Indus River Basin, Elsevier, 267-308, https://doi.org/10.1016/B978-0-12-812782-7.00013-8, 2019.

Ashraf, M., Ul Hassan, F.: Groundwater management in Balochistan, Pakistan: a case study of karez rehabilitation, Water Knowledge Note Washington, D.C., World Bank Group, http://documents.worldbank.org/curated/en/410961579803784924, 2020.

Ajmal, M., Waseem, M., Ahmad, W., and Kim, T. W.: Soil moisture dynamics with hydro-climatological parameters at different soil depths. Envir. Earth Sci., 75, 1-15, https://doi.org/10.1007/s12665-015-5021-3, 2016.

Deng, S., Tan, X., Liu, B., Yang, F., and Yan, T.: A reversal in global occurrences of flash drought around 2000 identified by rapid changes in the standardized evaporative stress ratio. Sci. Total Environ., 848, 157427, https://doi.org/10.1016/j.scitotenv.2022.157427, 2022.

Dimri, A. P., Kumar, D., Chopra, S., and Choudhary, A.: Indus River Basin: future climate and water budget, Int. J. Climatol., 39(1), 395-406, https://doi.org/10.1002/joc.5816, 2019.

Feng, J., Li, J., Xu, C. Y., Wang, Z., Zhang, Z., Wu, X., and Jiang, S.: Viewing soil moisture flash drought onset mechanism and their changes through XAI lens: A case study in eastern China, Water Resour. Res., 60, e2023WR036297, https://doi.org/10.1029/2023WR036297, 2024.

Li, J.Y., Wu, C.H., Xia, C.-A., Yeh, Pat, J.-F., Chen B., Lv, W.H., and Hu, B.X. 2022. A voxel-based three-dimensional framework for flash drought identification in space and time. Journal of Hydrology, 608: 127568.

Mahto, S. S., and Mishra, V.: Increasing risk of simultaneous occurrence of flash drought in major global croplands, Environ. Res. Lett., 18, 044044, http://doi.org/10.1088/1748-9326/acc8ed, 2023.

Otkin, J.A., Anderson, M.C., Hain, C., Svoboda, M., Johnson, D.K., Mueller, R., Tadesse, T., Wardlow, B.D., and Brown, J.F.: Assessing the evolution of soil moisture and vegetation conditions during the 2012 United States flash drought, Agric. For. Meteorol., 218219, 230–242, https://doi.org/10.1016/j.agrformet.2015.12.065, 2016.

Otkin, J.A., Zhong, Y., Hunt, E.D., Basara, J.B., Svoboda, M., Anderson, M.C., and Hain, C.: Assessing the Evolution of Soil Moisture and Vegetation Conditions during a Flash Drought–Flash Recovery Sequence over the South-Central United States, J. Hydrometeorol., 20, 549–562, https://doi.org/10.1175/jhm-d-18-0171.1, 2019.

Qing, Y., Wang, S., Ancell, B., and Yang, Z. L.: Accelerating flash droughts induced by the joint influence of soil moisture depletion and atmospheric aridity, Nat. Comm., 13, 1139, https://doi.org/10.1038/s41467-022-28752-4, 2022.

Waseem, M., Ahmad, I., Mujtaba, A., Tayyab, M., Si, C., Lü, H., and Dong, X.: Spatiotemporal dynamics of precipitation in southwest arid-agriculture zones of Pakistan, Sustainability, 12(6), 2305, https://doi.org/10.3390/su12062305, 2020.

---

## Author Comment (AC2)

**Spatiotemporal Evaluation of Vertical Dynamics Propagation of Flash Drought and Driving Mechanisms in the Indus Basin in South Asia (1970-2023)**

Tahira Khurshid[1], Qiongfang Li[1,2], Chuanhao Wu[2], Akif Rahim[3], Muhammad Shafeeque[4,5,6], Shanshui Yuan[2], Zia Ul Hassan[7], Junliang Jin[2,8,9]

**RC2 comments' response:**

1) Vertical Dynamics Propagation: The manuscript claims to investigate vertical dynamics propagation of flash drought. However, the current analysis merely compares the frequency and severity of flash drought events across different soil layers. There is no clear assessment of e.g. time-lag relationships or cross-layer dependencies that would explain vertical propagation.
Response
Thank you for your valuable comments. In this study, the vertical flash drought propagation process is represented by the downward movement of drying conditions through soil layers. Specifically, moisture deficits in the upper layer during an FD event induce deficits in the middle layer, leading to FD occurrence at that depth. We identified two types of the events i.e., simultaneous ('t') and subsequent ('t+1'), based on the criteria mentioned in section 2.4 (see L187-190). 't' events signify the quicker response of middle layer to upper layer, while in case of 't+1' events, middle layer response was observed at the difference of 1-timestep (i.e., 8-days).

Following your comments, we will include an analysis of the soil drying relationship between the upper and middle layers to explain the vertical propagation and discuss the underlying mechanisms of vertical flash drought propagation in the revised MS.

2) Limited study domain: The study is limited to the Indus Basin, which limits the generalizability of the findings. While this area provides valuable context, the broader applicability of the results remains unclear. Given that the study used GLDAS, a global dataset, the authors could easily extend their approach across a wider spatial domain.
Response
Thank you for your valuable comments. The Indus Basin has a wide range of climate types, with a humid- semi humid climate in the north and a semi-arid to arid climate in the south. Moreover, - this region is one of the most frequent flash drought prone areas in the world (Christian et al., 2021). Therefore, the Indus Basin is a representative area for studying the spatiotemporal evolution of flash drought, which can provide valuable scientific insights for other similar climate regions.

We fully agree with you that extending the study to a wider spatial domain can better demonstrate the broader applicability of the results. This prompts us to strive to collect more soil moisture data in future research and expand the study to continental or global scales to explore the vertical propagation mechanisms of flash drought under a wider range of climate types and different underlying surface characteristics. Following your comments, we will add more

explanations for choosing the Indus Basin as the research area for flash drought research in the revised MS.

3) Lack of Original Interpretation in the Discussion: The discussion is mostly revised on summarizing previous studies, with limited attention to the novel contributions of this work. The authors should highlight how their findings confirm/contrast, or advance the existing knowledge.

Response

Thank you for your valuable comments. Following your suggestions, we will enhance our discussion section to highlight how our findings confirm or advance the existing knowledge in the field by focusing on the underlying mechanisms of vertical flash drought propagation and the discrepancy in the characteristics of flash drought across different soil layers.

**References:**

Christian, J. I., Basara, J. B., Hunt, E. D., Otkin, J. A., Furtado, J. C., Mishra, V., ... & Randall, R. M. (2021). Global distribution, trends, and drivers of flash drought occurrence. Nature communications, 12(1), 6330.